# European Union Tools for the Sustainable Development of Border Regions

Florentina Chirodea [1,*] , Luminita Soproni [1,*] and Mihai Marian [2]

1 Department of International Relations and European Studies, University of Oradea, 410087 Oradea, Romania
2 Psychology Department, University of Oradea, 410087 Oradea, Romania; mmarian@uoradea.ro
* Correspondence: fchirodea@uoradea.ro (F.C.); lsoproni@uoradea.ro (L.S.)

**Abstract:** The European Union uses several instruments aimed at reducing disparities between regions and strengthening their competitiveness and sustainability. The border regions have a special relevance, given their position and characteristics, as well as the challenges faced by local actors. The European Commission has introduced the action "b-solutions" to tackle the specific border obstacles along European Union internal borders. The article aims to analyze the integration mechanisms at the micro-regional level, which is considered as a viable and sustainable solution for cross-border regions with resources which can be exploited to attract investment and generate wealth and well-being. The qualitative analysis method involved consulting thematic publications of the b-solution program and extracting data from the presentation sheets of accepted proposals published by the Association of European Border Regions. The collected data were processed according to criteria such as direction of action, types and causes of obstacles, solutions proposed by experts, etc. Addressing the legal and administrative obstacles that hamper cross-border flows proved to be a good initiative, with 120 cases selected. The solutions offered to the particular cases can also be replicated for other obstacles identified at the European Union's internal borders so that cooperation between border regions is intensified to the benefit of increasing European territorial, economic, and social cohesion.

**Keywords:** border region; cross-border partnership; sustainable development; b-solutions program

## 1. Introduction and Theoretic Background

Regionalization is an attractive option in the contemporary socio-economic landscape because it increases the capacity of states to attract foreign investments and allows for development through the use of common resources, which results in a reduction in certain adjustment costs and helps to overcome some political and administrative obstacles [1–5]. It thus supports the more facile integration of the countries involved into the global market.

Regionalization is defined differently by specialists depending on the relationship they establish between globalization and regionalization. Those that see regionalization as an opposed process to globalization consider that this removes the advantages of liberalization and allows regional partners to be privileged over others [6]. Other authors define a second perspective, which considers regionalization as a logical response to the effects created by globalization, as agreements between states are necessary for these to become stronger in international economic competition [1,7–9]. The group of specialists of the third line of analysis considers regionalization as a stimulus of globalization and therefore as a basic component of a deeper integration. The fact that states demonstrated readiness to participate in regional agreements proves that such agreements are complementary to the globalization process rather than attempts to replace it [10–13].

The new type of regionalism is a worldwide phenomenon that emerged in the 1980s as a result of the need to harmonize national policies with the view of achieving global economic integration [14]. It occurs in several areas of the world and continues the processes and mechanisms created by the old regionalism (manifested in the 1950s–1970s),

which mainly aimed to reduce or eliminate barriers to trade. The process was facilitated by the negotiations within the Uruguay Round of the General Agreement on Tariffs and Trade (GATT). At the same time, the new regionalism defines and develops new cooperation mechanisms and structures (Figure 1). If we refer to the actors involved in the regional process, we find that the nation-states played a central role in the case of old regionalism, while new regionalism involves actions and interactions of several players, such as regional and local authorities and civil society alongside the actors from businesses and non-governmental bodies [15–18]. In this sense, the European Union is considered a model of new regionalism due to its characteristics, which are seen as essential for deeper integration [8,19–22].

**Old model**
Government shapes economic development through economic policy decisions and incentives

**New model**
The economic development requires a process of cooperation, which involves public authorities at different levels, educational and research institutions, actors from the private sector

**Figure 1.** Shifting responsibilities for economic development. Source. Authors' own elaboration based on the theoretical background.

The new regionalism has two components. Its first direction concerns cross-border groupings established at the meso-regional (different states wishing to integrate their economic, political, or military activities) and macro-regional levels (trade blocs, regional organizations, or transcontinental networks). The decision of states to enter into a regional agreement is largely based on the balance between the costs and the economic benefits at stake. In addition, the size of a country influences both the decision concerning the association and its ability to use power in international relations [23]: small countries move towards regionalism because they believe that their relative position can be weakened if they remain isolated; at the same time, they aim to secure their position against the risks of disputes with major partners. On the other hand, large countries see regionalism as a way to increase their power and expand their influence in international negotiations.

If we look at the regional integration agreements made at the mezzo or macro-regional level [24–26], the problem that arises here relates to the fact that the main beneficiaries of these forms of integration are the developed states. Developing countries' governments therefore face both short-term and long-term challenges in increasing regional and global competitiveness, and traditional responses no longer offer viable solutions, hence the need for the second component of new regionalism: the micro-regional perspective.

In the context created by the mechanisms and processes of the new regionalism, the actions carried out at the sub-national level have become an integral part of the vocabulary related to development (especially in the case of the neo-liberal model in the northern hemisphere) [27,28], highlighting the relevance of the sub-national dimension of progress and development. Post-development theorists support the need for local objectives to be prioritized in the development of any policies, instead of including individuals and communities in programs and projects created for global purposes for which they do not have the necessary power. They envision and articulate development in different terms than the proponents of the global development concept [29,30], believing that actions with global scope and effects can be more effectively carried out at the local level. This theory is therefore about thinking and acting locally without excluding transnational alliances and networks [31,32].

This perspective fits into the theory of endogenous regional development [29,33,34], which is based on the internal development potential of the local community. The model prioritizes regional needs within national territory and capitalizes on territorial strategic

advantages that provide a competitive position to the region. Thus, the region must be able to guarantee the autonomy of processes aiming at economic growth through its own resources and through the advantages offered by the local specificity. In addition, the endogenous growth model considers interactions between regions [35–37]. Thus, endogenous factors are considered drivers of regional development and growth and are enhanced by the economic, social, or cultural interconnections achieved between or among neighboring regions through cooperative actions.

At the level of the European Union, the need to overcome the obstacles generated by the processes of deepening economic and political integration has led to increased interest in the field of regional development policy. Cohesion policy, being a horizontal policy, addresses areas considered strategic by the European Commission (education, employment, energy, the environment, the single market, research, and innovation) by financing the territorial development programs proposed by the member states, which are implemented at regional level by local authorities. The EU has created a series of territorial cooperation instruments within its Cohesion Policy. The New Cohesion Policy (2021–2027) continues and complements the objectives pursued during previous periods [38,39].

Processes aimed at European integration have led to the transformation of border regions from peripheral areas into areas of growth and development. The changes generated by the European integration process (increased productivity, reduction of transaction costs, intensification of intra-European trade, and increase in the number of jobs) had both positive outcomes (increased cross-border interactions) and negative outcomes (reduction in the number of jobs in the field of customs administration) in border regions [40]. In the context of the development model described above, border areas are often described as "laboratories" of European integration and cross-border cooperation, as they are hotspots with intense cross-border interactions. They are regions where the advantages of the single market are visible and where new ideas and solutions can be tested for the first time and analyzed at a small scale [41–45].

The areas bounded by the internal borders of the EU, which are more or less open, represent interesting territories for researchers because three major changes can be analyzed here [46]: (1) the increase in cross-border trade and service flows, along with the increase in the international mobility of the workforce, as a result of the effects of European integration; (2) the expansion of transport networks, utilities and public services and the emergence of new models of economic activities, as a result of investments in transnational infrastructure; (3) strengthening cooperation between communities located on the two sides of the border and multiplying cross-border development initiatives, by standardizing legal and administrative procedures. Moreover, the border regions delimited by the old border that divided Western and Eastern Europe, also called "little Europe", seem to be the most suitable areas for analyzing opportunities for political, economic, cultural, environmental, and social welfare action [47].

The emergence and development of cooperative relations between border regions have been supported by a series of European Commission initiatives. Most of these have been transposed into EU legislation [48]. Since 1990, the most important instrument for the implementation of the European Union's Cohesion Policy in border areas has undoubtedly been the European Cross Border Cooperation program (Interreg), which aims to encourage economic growth in border areas, having among its objectives the acceleration of regional development [49]. The program was created to stimulate cooperation between institutions and communities located on both sides of the border, through the development of cross-border socio-economic centers with common development strategies (Euroregions) [50–52]. Interregional cooperation projects are included in the first pillar of the Interreg program, which aims to improve the exchange of experience and the sharing of common practices, as well as the preparation of action plans for the integration and implementation of good practices within regional development policies [42].

In this context, for the purposes of this work, we have formulated the following research hypothesis: the integration mechanisms at the micro-regional level (which form

the second direction of the new regionalism) appear as a viable solution for cross-border states or regions that have resources which can be exploited in order to attract investment and generate wealth and well-being at their internal level.

The research questions subsumed to this objective are:

1.  Is the b-solutions program an effective mechanism of European integration through the instruments offered to the beneficiary regions?
2.  Have the obstacles identified in the selected proposals been eliminated by applying the solutions proposed by the experts?
3.  Is the lack or poor functioning of institutional cooperation one of the important sources of obstacles?

The paper is organized in such a way as to provide a contextualization of the assumed research hypothesis by creating a theoretical framework that explains the premises and specificity of the new regionalism, seen as the foundation for the relevance of the sub-national dimension of progress and development (expressed starting from the theory of endogenous regional development). At the same time, regionalism is seen as context for the cross-border actions and programs analyzed. One section of the paper includes a review of the process that resulted in the financing of the b-solutions initiative by the European Commission, with a presentation of the objectives and directions of action related to the program. Cases associated with the process, as well as its results, are described in a separate section. The last part of the paper provides answers and validations/invalidations for the hypothesis and the research questions posed.

## 2. Materials and Methods

The study was conducted between 2021 and 2023, starting with the qualitative analysis of data made public by the Association of European Border Regions (AEBR). In the first part of the research, the three thematic publications developed by the AEBR in 2021 were consulted regarding obstacles and solutions to cross-border cooperation in the EU. In the second part of our research, focused on the fourth direction of the program (institutional cooperation), data were extracted from the presentation sheets of the proposals accepted for analysis, published in the annex to *B-solutions: Solving Border Obstacles. A Compendium of 43 Cases,* respectively in *B-solutions: Solving Border Obstacles. A Compendium 2020–2021.*

In addition, for an in-depth analysis of the effectiveness of the b-solutions initiative, we consulted two studies developed by groups of researchers led by Eduardo Medeiros, published in 2021 and 2023, which address the issue of the b-solutions program. The diagram shown in Figure 2 schematically shows the analysis process.

The collected data was processed according to a set of criteria such as: the direction of action proposed by the program; the types of obstacles (legal, administrative), the causes that led to the emergence of obstacles; the solutions proposed by experts (legislative, capacity building and administrative coordination, transversal solutions). In line with the standard criteria of exploratory qualitative analysis, we explained how the b-solutions initiative worked and its impact was. On the other hand, we have considered that the method can suggest possible relationships, causes, effects, or dynamic processes in our research area. Additionally, in the qualitative analysis we have targeted a set of keywords (e.g., direction for action, obstacles, causes, policy area, etc.) to facilitate the identification of relevant information.

For three directions of action of the program (cross-border public services, the labor market and education, the implementation of the Green Deal), we made a synthesis of the results published in the three publications of the AEBR mentioned above and referred briefly to the most important achievements in each field. For the fourth line of action, institutional cooperation, which is not the subject of an analysis by AEBR specialists or other researchers, we created a database for the 24 selected proposals, in which information was entered based on the following criteria: local actors who submitted the proposal; policy area; the indicated obstacle; type of obstacle (legal, administrative); causes of obstacles;

the proposed solutions. The data were analyzed qualitatively so that along with those presented synthetically, they can be used in a deep analysis of the b-solutions program.

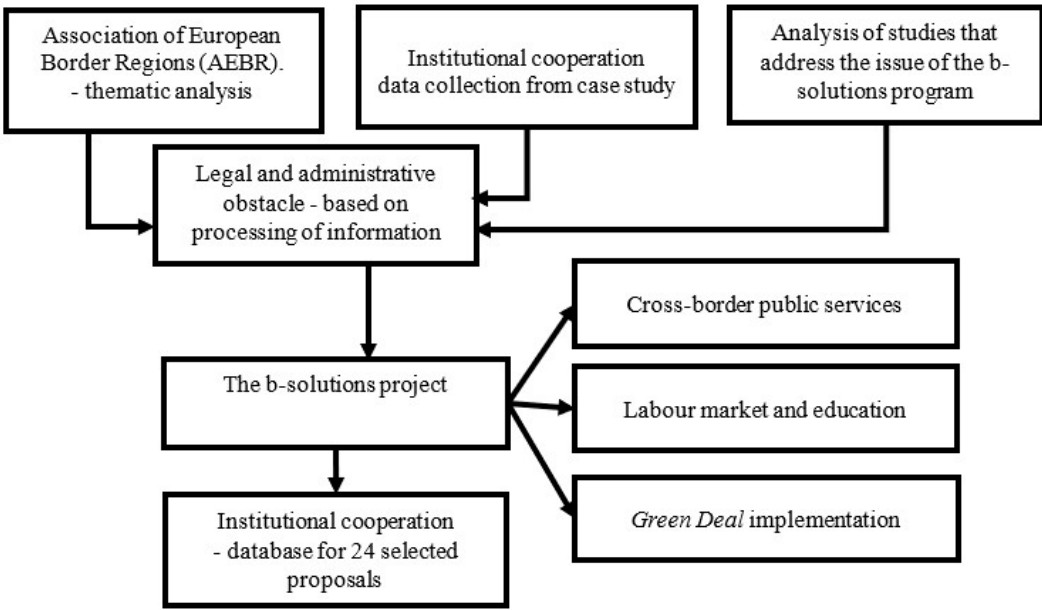

**Figure 2.** Data analysis mechanism used for the b-solutions initiative. Source. Authors' own elaboration.

## 3. Results

Since its launch in 2017, the b-solutions initiative, through the experts involved, has identified and promoted sustainable ways to reduce obstacles at the EU's internal borders, including neighboring EFTA countries. The program provided the opportunity to effectively test ways to overcome barriers and replicate solutions to achieve full cross-border cooperation [53]. Initially described as an EU pilot action through which 10 projects and 33 case studies were selected, where divergences between national and European legislation or incompatible administrative procedures are indicated, the initiative highlighted the European Commission's interest in "collecting practical, feasible, comprehensible, viable solutions, aimed at reducing the identified barriers" [54]. The thematic areas of the 10 pilot actions (labor market, health, public passenger transport, multilingualism, and institutional cooperation) and those in which the case studies are grouped (e-government, labor market, records and databases, health, information services, institutional cooperation, multilingualism, and transport) highlighted the fact that most cross-border obstacles are based on differences in the legislation of the member states, followed by incompatibility or duplication of administrative procedures. Their reduction or elimination requires actions that especially involve the public administrations at different decision-making levels in the member states but also other local actors. After the first phase of the b-solutions program, AEBR published a compendium with the cases analyzed and the resulting solutions, offering the opportunity to replicate them in the cases of other internal borders or introduce them into EU legislation [55].

In a complex analysis carried out in 2021, a group of researchers led by Eduardo Medeiros [56] showed that there is an unbalanced geographical distribution of pilot actions and cases accepted for analysis, with most projects (40%) coming from the Benelux and from the border of France with Germany. The explanation offered by the researchers refers to the cross-border institutional maturity, the intensity of cross-border flows, the deep integration of the regions in these areas, and the fact that the borders in northwest Europe are those with the most legal and administrative obstacles. Unlike the pilot projects, the 33 cases selected in the first phase of the project "are better distributed along the European borders", as Medeiros and his collaborators state (70% of the cases are concentrated in the border areas between the Benelux countries–France–Germany–Spain–Portugal), and

the thematic area covered is considered to be much more relevant to the objectives of the program. In general, the result of this first phase consists in the identification of concrete solutions to reduce the various cross-border barriers in transport, trade, health, education, language, taxation, environment, cartage, etc., the implementation of the solutions not being a priority of the program. However, the actors involved in the development of pilot proposals and cases have started to take the first steps in involving different authorities or entities in a common process of reducing or eliminating obstacles [56].

The second, third, and fourth phases of the b-solutions program, carried out in the period from 2019–2021, meant the acceptance of another 47 cases for which AEBR experts offered solutions to reduce legislative and administrative obstacles. The Association's specialists have published a series of summaries of the data collected in the analysis sheets of projects and cases from the 4 years of implementation of the b-solution program, which cover only three of the four major directions of action of the program: cross-border public services; labor market and education; implementation of the Green Deal. A summary of the published results is presented synthetically in Table 1.

The data collected by AEBR experts for the field of cross-border public services highlighted the presence of obstacles of a legal and administrative nature in the following thematic areas: citizenship, justice, public security; civil protection, disaster management; communication and information of society; education; environment protection; health; social inclusion; labor market; spatial planning, culture; transport [57]. In the absence of common methodologies, collaboration protocols and coordination structures, the attempt of local authorities to offer or stimulate certain services is cumbersome or delayed, with consequences on the achievement of the European objective of being as close as possible to citizens in all regions [57].

Regarding the field of education and the labor market, the AEBR specialists say that due to the multitude of unclear rules that must be respected and the many existing administrative models in the Member States, coordination across internal borders is a complex process hampered by difficulties in areas such as social security; diplomas and certificates recognition; access to training and education; business opportunities; trade; and professional or educational status of non-EU nationals [58]. The solutions offered in this field are general and at the level of good practices, they must be customized according to the specific legislation and national practices by the competent actors in education and the labor market, with local public authorities having the task of applying common actions and to engage the opportunities of the single market [58].

For the implementation of the Green Deal strategy adopted by the EU in 2019, which had a major role in regional development [59], local actors from the regions on the internal borders of the EU have noticed difficulties in the following areas: clean, cheap and safe energy; clean and circular economy; energy-efficient buildings with low resource consumption; sustainable mobility; environmentally friendly food system; conservation of ecosystems and biodiversity; and zero pollution [60]. The collected data show that border regions have to deal with the same obstacles in carrying out sustainable common actions, and the signing of cross-border agreements between local or national authorities for the implementation of green projects is the most often offered solution. AEBR experts also recommend increasing access to training courses through the INTERREG program, harmonizing green initiatives and involving EGTC-type structures as potential promotors of cross-border initiatives aimed at sustainable development [60].

**Table 1.** Overview of AEBR publications that refer to the results of implementing the b-solution program for three directions of action.

| Direction for Action | Obstacles Identified | | Causes | Other Causes | Solutions Proposed | | |
|---|---|---|---|---|---|---|---|
| | Legal | Administrative | | | Legal | To Strengthen Administrative Capacity and Coordination | Transversal |
| Cross-border public services | - Local and regional levels of government do not have legal powers to provide or operate services<br>- The legislation does not allow the automatic recognition of diplomas issued by another EU member state<br>- Different national rules governing services<br>- Technical standards/requirements for the operation of services regulated differently at national level<br>- The legislation regarding the provision of services does not take into account the cross-border dimension;<br>- Conflicting transposition of EU legislation in cross-border regions. | - Long procedures for the recognition of diplomas<br>- Absence of common administrative mechanisms to facilitate the operationalization of services<br>- Lack of knowledge of the decision-makers regarding the enabling legislative framework to regulate certain services<br>- Different approaches in data collection as a preliminary step towards the implementation of public services<br>- The presence of many actors with different levels of administrative skills in certain fields. | - The failure to take into account the specificity of cross-border territories, which creates gaps where inconsistencies are generated | - Complex and burdensome administrative procedures for coordinating public services across national borders<br>- Lack of horizontal coordination between responsible bodies<br>- Lack of knowledge of the decision-makers regarding the legal or administrative framework<br>- Incompatibility or lack of provisions at various legislative levels (sub-national, national, European). | - Changes in legislation<br>- Revision or improvement of the existing one<br>- Creating an ad hoc legislative framework. | - Creation of new coordination structures<br>- The development of ad hoc conventions. | - Specific strategies for increasing coordination among the actors involved<br>- Training courses to increase knowledge in new cooperation schemes<br>- The complementarity of EU aid (e.g., INTERREG)<br>- The establishment of cross-border structures creating ad hoc plans<br>- development of protocols and conventions. |

**Table 1.** *Cont.*

| Direction for Action | Obstacles Identified | | Causes | Other Causes | Solutions Proposed | | Transversal |
|---|---|---|---|---|---|---|---|
| | **Legal** | **Administrative** | **Causes** | **Other Causes** | **Legal** | **To Strengthen Administrative Capacity and Coordination** | **Transversal** |
| Labor market and education | - The EU framework contains general provisions, leaving room for interpretation<br>- The provisions in force are not adapted to the complexity of the cross-border context<br>- The national provisions regarding contracts, taxation and financing are not aligned<br>- The regulations in force do not provide automatic recognition of diplomas/certificates<br>- The provisions regulating new fields/working and living conditions are insufficient or outdated. | - The presence of complex/unclear bureaucratic procedures<br>- The absence of common administrative mechanisms<br>- The existence of different administrative protocols and approaches<br>- Lack of knowledge as regards the frameworks in force that are already facilitators. | - Lack of provisions to regulate different issues<br>- Lack of knowledge about the already existing framework that facilitates the provisions/tools offered by the legal/administrative framework in force<br>- Divergent or inconsistent regulations at European, national or sub-national level. | - Complex and burdensome administrative procedures on both sides of the border<br>- Lack of administrative coordination between competent actors from two or more neighboring countries. | - Amending or improving existing legislation at European level<br>- Revising or updating the current provisions on one or both sides of the border<br>- Creating an ad hoc legal framework. | - Creating ad hoc plans<br>- Development of protocols and conventions. | |

**Table 1.** *Cont.*

| Direction for Action | Obstacles Identified | | Causes | Other Causes | Solutions Proposed | | |
|---|---|---|---|---|---|---|---|
| | Legal | Administrative | | | Legal | To Strengthen Administrative Capacity and Coordination | Transversal |
| Green Deal implementation | - Creation and management of common infrastructures<br>- Divergent national rules on infrastructure design and construction approvals<br>- Inconsistent legal powers of spatial planning<br>- Different regulations for the necessary technical requirements;<br>- Lack of revised legislation<br>- Lack of specific provisions regarding the cross-border dimension<br>- Absence of standardized European norms. | - Different references for spatial data needed for mapping and data collection<br>- Different technical standards for environmental management criteria<br>- The absence of an ad hoc cross-border structure or entity, responsible for the coordination of nature reserves<br>- The absence of a common mechanism for regulating data exchange. | - Inconsistencies in the legal frameworks related to green policies in place in neighboring Member States<br>- Member States' exclusive competence on certain matters that regulate actions, infrastructure and projects implementing the Green Deal<br>- National laws have not transposed the most recent EU legal framework. | - Diverging practices<br>- or technical features of specific actions hamper the completion of cross-border actions<br>- The lack of horizontal.<br>- cooperation among the stakeholders involved in a specific project or action. | - Amending and improving existing legislation at European level<br>- Harmonization at the supranational level<br>- Reviewing the current provisions on both sides of the border,<br>- creating an ad hoc legal framework. | - Establishing a common management structure<br>- Creating a single, unified command point<br>- Harmonization of environmental management data sets, methods and technical standards. | - The creation of a specific consortium for relevant actors on both sides of the border<br>- Information actions<br>- The training of local actors engaged in specific projects. |

Note: Own processing of the data presented by the AEBR in publications related to the implementation of the b-solution program [57–59].

For the fourth direction of action of the b-solutions initiative—institutional cooperation—AEBR has not yet developed a synthesis of the cases. Therefore, in what follows, we will carry out our own analysis of the information collected from the program's website. The importance of institutions for the success of reforms is widely recognized by the academic community, Dimitrios Zikos defining them "as systems of established and embedded social rules that structure social interactions. Institutional arrangements influence governance structures, shape the economy and affect public awareness and civic engagement". In this context, institutional cooperation is one of the tools often used to solve acute problems, in particular legal and administrative obstacles, at the internal borders of the EU. Zikos emphasizes the fact that this cooperation cannot exist outside of a framework of "formal rules", which generates a feeling of trust that the institutions are able to "provide expectations, stability, meaning essential to human existence, coordination, regularize life, support values and produce and protect interests" [61]. For their part, Ezers and Naglis-Liepa see institutional cooperation as a process carried out both horizontally and vertically, "opposed to competition", and demonstrate that its success influences regional development and economic progress. The two Latvian researchers emphasize that at least three of the principles of good governance "directly underpin the significance of institutional cooperation: openness—institutions have to be more open and actively communicate with one another; participation—the effectiveness of institutions is directly dependent on how successful the participation of the other ones is; coherence - cooperation among regional institutions have to be coordinated" [62]. It is also well known that the European Union has developed a wide spectrum of institutional cooperation mechanisms to achieve common objectives, which impose consultation rules and conflict management; provide predictability in decision-making processes; and impose certain obligations and constraints on different institutional actors [63]. The b-solution initiative contributes to the creation of such mechanisms, this time customized and much better anchored in the complex context of cross-border cooperation.

In the first phase of the program, 4 projects and 12 case studies were selected within the institutional cooperation direction of action, and in the period from 2020–2021, another 8 cases were analyzed by AEBR experts. The 24 selected proposals were registered in the following thematic areas: cross-border mobility; economy; sustainable management of rural areas; groundwater cross-border management; institutional cross-border cooperation; environmental management and circular economy; cross-border transport and traffic regulations; social and medical care; and protection and education of children.

The spatial distribution of the 24 proposals accepted for analysis in the direction of institutional cooperation, presented in Figure 3, is to some extent balanced in the EU space. It is observed, once again, to be a concentration of obstacles at the borders of the states that form the European core (Belgium, the Netherlands, Germany, France, and Luxembourg) but also a desire to eliminate difficulties on the part of the states of Central and Eastern Europe (Slovenia, Croatia, Hungary, Czech Republic, Poland, Lithuania, Latvia, Slovakia, Romania, and Bulgaria). We must also note the efforts of Spain and Portugal to make cooperation at their common border more efficient (see Figure 4).

The multitude and complexity of institutions from different levels of government; different institutional and legal cultures from one member state to another; the activity of cross-border actors (e.g., EGTC); the lack of bilateral agreements, ad hoc agreements, or unwritten customary procedures, customs control, and operations; and lack of coordination between stakeholders and actors are part of the factors that make it difficult to implement cross-border cooperation projects. The legal and administrative obstacles mentioned in the 24 selected proposals are presented in Table 2, which also highlights their distribution by identified thematic areas.

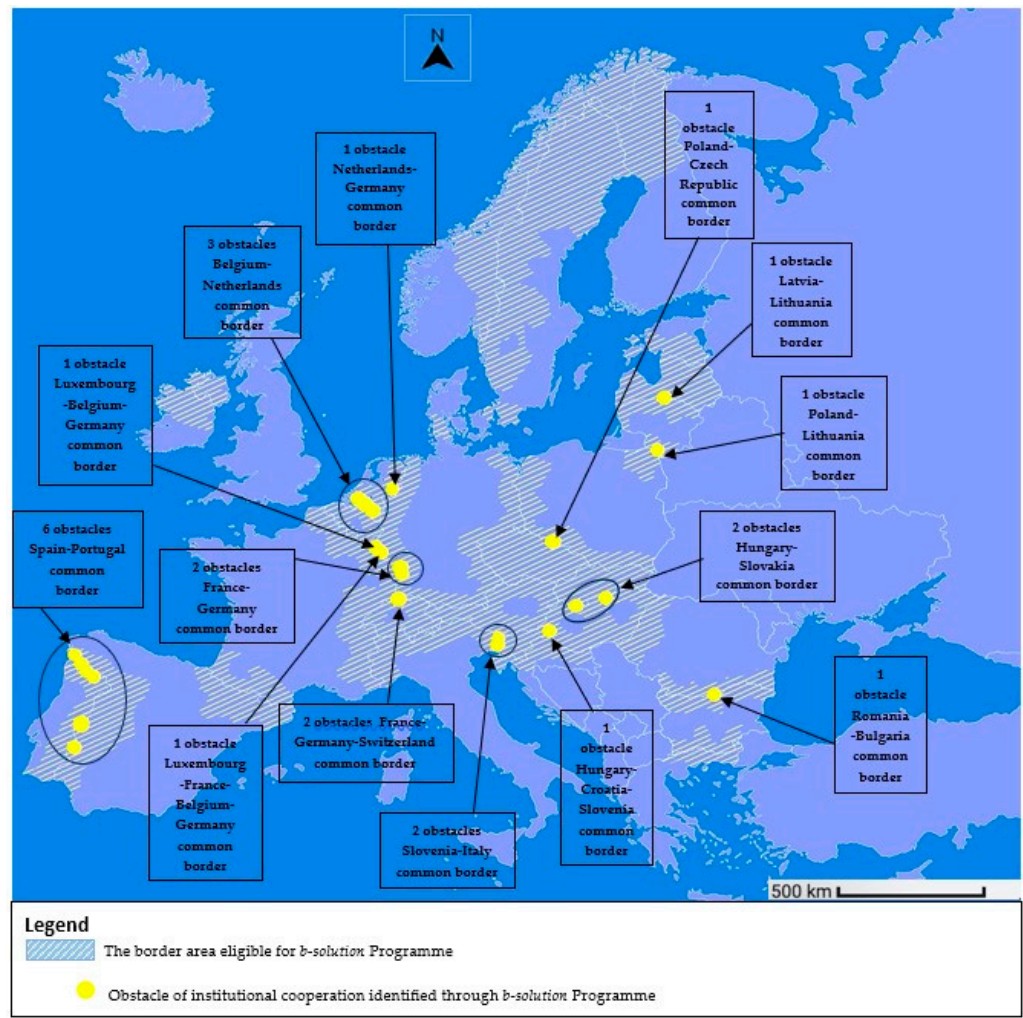

**Figure 3.** Distribution of institutional cooperation obstacles identified within the b-solutions program. Source: Our own elaboration, based on data found in Association of European Border Regions and European Union. *B-solutions: Solving Border Obstacles. A Compendium of 43 Cases*, Publications Office of the European Union: Luxembourg, Luxembourg, 2020 and Association of European Border Regions and European Union. *B-solutions: Solving Border Obstacles. A Compendium 2020–2021*, Publications Office of the European Union: Luxembourg, Luxembourg, 2021.

**Table 2.** Distribution of cross-border obstacles by thematic areas.

| Policy Area | Common Obstacles |
| --- | --- |
| Cross-border mobility | Different, divergent, and complex national administrative procedures; lack of clarity in information; language gaps; lack of reciprocity; lack of established rules in the field of right of access in the enclave. |
| Economy | Complex national customs procedures that make difficult the interoperability, with consequences upon the cross-border commerce; different legal frameworks in tourism; different national laws on taxation in case of customs free zone. |
| Sustainable management of rural areas | Different administrative systems for the projects application and implementation. |
| Groundwater cross-border management | Harmonization of data and methodologies. |

**Table 2.** *Cont.*

| Policy Area | Common Obstacles |
|---|---|
| Institutional cross-border cooperation | Tax differences and non-harmonized procedures; lack of clarity in the application of existing national and European regulations; unsureness regarding the rights of the employee and the obligations of the employer(s); lack of a financial support for EGTCs and diverging national legal frameworks; lack of recognition of EGTCs as legal entities in the Member States; different legal, organizational and technical principles regarding standardization of spatial data; uncertainty in administration of a complex situation; divergence in the implementation of the Bologna standards. |
| Environmental management and circular economy | National regulations require many changes to adapt to the latest EU legal provisions, because of that—legal and administrative requirements have grown exponentially; the non-harmonized rules on public procurement. |
| Social and medical care | Application of the national law in relation to the European Regulations about emergency assistance, and health insurance registration and inadequate division of responsibilities in difficult case. |
| Cross-border transport and traffic regulation | Different traffic regulations and difficulties in the coordination regarding transportation of resources for industrial processes. |
| Children protection and education | Complexity of the national legal and administrative requirements regarding protecting the interests of minors abroad; lack of common procedures between administrative departments and social services. |

Note: Own processing of the data presented by the AEBR in publications related to the implementation of the b-solution program [64,65].

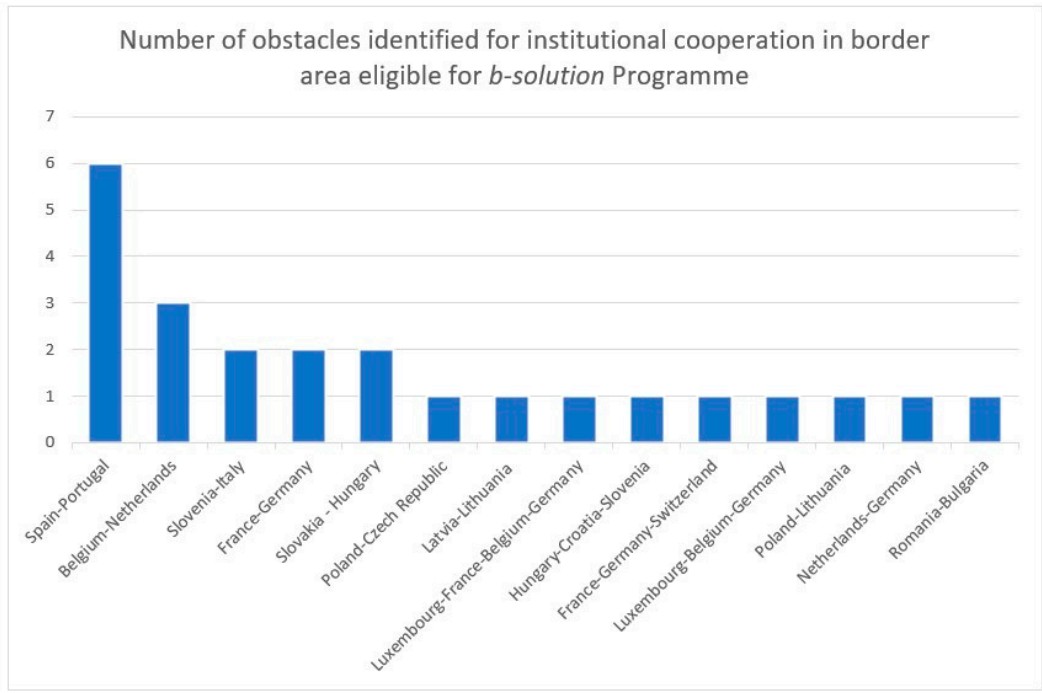

**Figure 4.** Number of obstacles identified for institutional cooperation in border area eligible for the b-solution program. Source: Our own elaboration, based on data from the Association of European Border Regions and European Union. *B-solutions: Solving Border Obstacles. A Compendium of 43 Cases*, Publications Office of the European Union: Luxembourg, Luxembourg, 2020 and Association of European Border Regions and European Union. *B-solutions: Solving Border Obstacles. A Compendium 2020–2021*, Publications Office of the European Union: Luxembourg, Luxembourg, 2021.

Analyzing the legal and administrative obstacles quantitatively, we observe the prevalence of those related to the different ways in which laws, regulations, and procedures are applied. At the same time, the differences between national legislative frameworks

can lead to the difficulty or even the blocking of institutional cooperation among different local actors located on different sides of the borders. In addition, the transposition of EU directives into national legislation in a different way leads to confusion and a lack of clarity in the application of national or European norms, the consequence being the multitude of types of administrative obstacles reported in cross-border cooperation. On the other hand, these obstacles, whether legislative or administrative, are also based on the lack of institutional cooperation between the local authorities located on either side of the border or the national authorities that would generate coordinated procedures, coordination between stakeholders and actors, interoperability, digitization of public administrations, forms developed in at least three languages (one of which is widely used), ad hoc agreements, and usually unwritten procedures. The b-solution program, through its objectives, addresses these obstacles, and the solutions proposed by AEBR experts are in line with these objectives, as can be seen from Table 3.

**Table 3.** The solutions identified by AEBR experts for the 24 cases selected for institutional cooperation.

| Objective of the b-Solution Program | Proposed Solutions | | |
|---|---|---|---|
| | Legal | Enhanced Administrative Capacity and Coordination | Cross-Cutting Solutions |
| O1. Mitigate cross-border obstacles which are caused by a lack of coherence, inconsistencies or overlapping between legal provisions or administrative procedures on each side of the border but, particularly, because applicable European, national or regional/local legislation does not consider the specificity of cross-border interactions | - Adapting and amending national provisions to respond to the exceptional context of a cross-border area; <br> - Taking into account the existing national legal framework for cross-border cooperation of enforcement authorities; <br> - Application of European regulations. | | - Lobbying administrations with the objective of simplifying procedures for citizens; <br> - The organization of dedicated trainings in bilingual legal assistance centers to facilitate the simultaneous fulfilment of legal procurement obligations. |
| O2. Increase the understanding of every specific obstacle among key stakeholders on both sides of the border at local, regional, national and EU levels. | - Drawing up an action protocol between the actors involved with the support of legal experts. | - Cooperation between national governments and European institutions and organizations. | - Creating an inventory of administrative obstacles faced by border residents. |
| O3. Promote sustainable methods to solve cross-border obstacles through innovative proposals to inform further cross-border development and implementation by public authorities or through EU instruments. | - Modify the LEADER scheme to the entire EU area. | - Transfer the submission responsibility and accountability for cross-border LEADER projects to one managing authority. | - Formalizing cross-border cooperation by establishing a local version of the European EGTCs. |
| O4. Involve public bodies committed to jointly fostering, designing, and agreeing on feasible solutions to reduce cross-border barrier effects. | - Updating and adapting existing bilateral agreements to current needs. | - Establishment of bilateral agreement. | - Coordinate local development strategies. |

**Table 3.** *Cont.*

| Objective of the b-Solution Program | Proposed Solutions | | |
|---|---|---|---|
| | Legal | Enhanced Administrative Capacity and Coordination | Cross-Cutting Solutions |
| O5. Stimulate an increased exchange of information and mutual engagement between the variety of administration levels in border areas to make possible the generation of joint initiatives involving multi-level governance across borders. | | - The unification of digital platforms for multimodal logistics integration;<br>- The IT specialists involved in the customs offices communicate closely to achieve the goals of interoperability and coordination. | - Information and knowledge sharing, structured consultation, organizing forums and events. |
| O6. Foster the replication of the solutions found. | | - Harmonize the reference datasets, produce a digital terrain model. | - Design cross-border coordination points under the European Cross-Border Mechanism (ECBM) |

Note: Own processing of the data presented by the AEBR in publications related to the implementation of the b-solution program [64,65].

## 4. Discussion

The new regionalism provides the appropriate framework for the fulfilment of the endogenous regional development model, as the actors involved in cooperation actions (states and regions) act strategically, pursuing the balance between the economic costs and the benefits at stake. Thus, new cross-border cooperation mechanisms and structures are generated and implicate the involvement of public authorities, educational and research institutions, and private sector companies on the playing board of regional competitiveness. The actions of local actors lead to the capitalization of territorial strategic advantages that provide a competitive position to the region. In this context, endogenous factors become drivers of regional development and growth, enhanced by the institutional, economic, social or cultural interconnections made between neighboring partner regions.

Cross-border areas have become laboratories of European integration and cohesion, as they are hot spots where multiple intense interactions are carried out. They are regions where the advantages of the single market can be observed and where new ideas and solutions can be analyzed on a small scale or be tested for the first time. The b-solutions program, through the general objective assumed, represents an effective way to promote sustainable solutions for legal or administrative obstacles that prevent or hinder cross-border cooperation. Thus, during its development, the program identified and promoted sustainable solutions to reduce legislative and administrative obstacles at the borders of the EU, including neighboring EFTA countries, providing the opportunity to effectively test ways to overcome obstacles and replicate solutions to achieve full cross-border cooperation.

Our study complements the three reports published by AEBR and other studies that address the subject with an analysis of the fourth line of action—institutional cooperation— thus contributing to the evaluation of the b-solution program. Therefore, the analysis of the data collected and processed has shown that our research objective and the starting hypotheses are mostly verified: integration mechanisms at the micro-regional level represent a viable solution for cross-border states or regions to ensure wealth and well-being within their territory.

In a special report published in 2021, the European Court of Auditors found that Interreg-type cooperation programs can only partially respond to cross-border challenges,

and due to insufficient resources, it is necessary to direct funding where the added value is the highest. The Court found that in most cases, the cooperation between the partners was limited to the presentation of a joint proposal for the purpose of financing the investments. In addition, most of the cross-border challenges identified are generated by "legal obstacles, in relation to the legislative frameworks at the EU, national or regional level, the rest being administrative" [66].

In this context, financing the b-solutions initiative is a necessary decision for the proper functioning of integration processes at the micro-regional level. The way in which this mechanism was conceived led to the identification of numerous legal and administrative obstacles, for which AEBR experts were able to offer short-, medium-, or long-term solutions (customized solutions for each particular case). They confirm Research Question 1 (the b-solutions program is an effective mechanism of European integration through the instruments offered to the beneficiary regions) and refer to the following aspects: (1) modifying/updating/adapting national and European legislation to respond to the complex conditions in the border regions; (2) the adoption of an ad hoc legislative framework and the development of bilateral conventions/protocols; (3) establishment of common management structures or a single unified command point; (4) harmonization of methodologies and technical standards; (5) creation of consortia for relevant actors on both sides of the border; (6) carrying out information actions; and (7) training of local actors through projects financed by Interreg.

The European Court of Auditors estimates that if these solutions were implemented and 20% of the existing obstacles to cross-border cooperation were removed, border regions would register a 2% increase in GDP and create more than 1 million additional jobs [65]. Therefore, the effectiveness of the initiative is argued by the forecasted economic indicators, but the responsibility of the approaches belongs to all administrative levels (European, national, regional, and local), which partially confirms Research Question 2 (the obstacles noticed in the selected proposals were eliminated by applying the solutions proposed by experts).

In order to avoid cumbersome legislative procedures, AEBR experts proposed the signing of conventions, agreements, and protocols between local authorities on both sides of the borders. In addition to these short-term solutions, there are also possibilities that require an average implementation period as they aim to create common local management structures or unified command points. The application of all solutions depends, again, on the will of public authorities to attract investment for the creation of wealth and welfare in the border regions.

Another advantage of this initiative is the fact that by publishing the files of the accepted proposals and the results obtained in the period from 2018–2021, any solution proposed by the experts can be replicated to minimize a similar obstacle at the internal borders of the EU. The decision to continue the program in the 2021–2027 budget exercise demonstrates the usefulness of this mechanism, whose effectiveness can only be proven after these data are collected.

Local public actors are the ones that must apply both national and European legislation at sub-national levels and ensure, from an administrative point of view, that the procedures are effective, flexible, and easy to follow and that they do not overburden the daily lives of citizens. At the same time, development strategies and policies place them at the center of growth mechanisms that relate to living standards and life quality. In the case of actors living in border areas, these challenges are felt even more acutely due to the absence of administrative capacity, investments and the cross-border context in which they perform. Cross-border cooperation instruments provide numerous cooperation opportunities for local authorities on both sides of the European borders. However, the b-solutions program has highlighted the fact that efficient collaboration among authorities (at all levels) might generate "openness and dilution of barriers among communities and institutions" [67]. The limited number of selected proposals in the direction of institutional cooperation allows us to state that Research Question 3 (the absence of or the defective functioning of institutional

cooperation is an importance source of generated obstacles) is confirmed to a lesser extent. Following the analysis of the typology of the registered obstacles, one can see that they fall within the same thematic areas as those observed within the first three directions of actions of the program. Moreover, as regards the particularized solutions provided by AEBR experts, it appears that all 90 matters share common factors that generate the most obstacles—lacks of information, communication, and cooperation between sub-national or national public authorities.

## 5. Conclusions

The b-solution program, with its current version b-solution 2.0, is a new tool through which the European Commission can increase cross-border cooperation at the internal borders of the EU. Tackling the legal and administrative obstacles that hamper cross-border flows and the daily lives of European citizens proved to be a good initiative, with 120 cases selected for analysis.

The solutions offered to the cases studied can also be replicated for other legal or administrative obstacles identified at the EU's internal borders so that cooperation between border regions is intensified and sustainable, to the benefit of increasing European territorial, economic, and social cohesion.

The main theoretical contribution of this study consists in formulating the hypothesis according to which the micro regional integration constitutes a favorable context for boosting up cross-border cooperation and provides the tools with the help of which the local actors from the regions close to the internal borders of the EU contribute to a deeper European integration.

The study highlights, through a qualitative analysis of the data published by AEBR, certain types of border obstacles that generate restrictions in micro-regional cooperation but also the solutions proposed to eliminate these blockages. The newest tool available to the local actors, the b-solutions program, validates through the proposed objectives and the results obtained the formulated research hypothesis, demonstrating that the elimination of legislative and administrative obstacles is one of the mechanisms that can support deeper European integration.

The proposed mechanism and solutions pave the way for cooperation also at the level of other border regions of the European Union. In this sense, we believe that the paper offers a model of analysis and work for regional public authorities looking for viable and sustainable solutions for the development of cross-border micro-regions.

From 2022, the b-solution initiative has continued with version 2.0 and will still remain in our evaluation for an even wider validation of our research hypothesis, especially since the eligibility area of the program has been extended to maritime border areas and common borders with countries involved in pre-accession assistance. It is also interesting to investigate how the solutions offered to solve legislative and administrative obstacles contribute to boosting up other cross-border cooperation programs (i.e., Interreg).

**Author Contributions:** Conceptualization, L.S.; methodology, M.M.; software, F.C.; validation, L.S., F.C. and M.M.; formal analysis, M.M.; investigation, F.C.; resources, L.S.; data curation, F.C.; writing—original draft preparation, L.S. and F.C.; writing—review and editing, M.M.; visualization, L.S., F.C., and M.M.; supervision, L.S., F.C. and M.M.; project administration, L.S.; funding acquisition, F.C. All authors have read and agreed to the published version of the manuscript.

**Funding:** This research received no external funding.

**Institutional Review Board Statement:** Not applicable.

**Informed Consent Statement:** Not applicable.

**Data Availability Statement:** The data presented in this study are available on request from the corresponding author. The data are not publicly available due to privacy considerations.

**Conflicts of Interest:** The authors declare no conflicts of interest.

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
