# Peer review of "European Union Tools for the Sustainable Development of Border Regions"

_sustainability, doi:10.3390/su16010388_

Round 1
Reviewer 1 Report
Comments and Suggestions for Authors
General comment: This is a study with an interesting and topical topic and objective, well-structured, but lacking depth in several sections to offer any significant original contribution.
General information: The source or authorship of the figures should be indicated.
Abstract: Clear and concise. The meaning of the acronym AEBR should be expressed.
1. Introduction and background:
- Lines 39-41 discuss the beginning of the new regionalism. Is this worldwide? Or more specifically in Europe, within the framework of the European Economic Community?
- A brief definition of the territorial scale of the mesoregional and macroregional levels would be helpful. Perhaps a footnote would suffice.
- More background on innovation in cross-border regional systems is needed. For example: Makkonen, T., Weidenfeld, A., & Williams, A. M. (2017). Cross‐border regional innovation system integration: An analytical framework. Tijdschrift voor economische en sociale geografie, 108(6), 805-820.
2. Materials and methods:
- It is essential to clarify some things in this section. Was any tool used to analyze the information? Is it a content analysis? What measures were taken to ensure the validity and reliability of the qualitative results?
- It should also be made clear which cases were worked with. Generating a map would facilitate the task.
3. Results:
- The cartography of Figure 3 needs improvement. A legend should be included that explains the elements incorporated (patches and points), and labels for the countries and regions of interest should be added. In addition, a legend and geographic north should be incorporated.
- If reference is made to a quantification of the results by thematic categories (314-316), it would be helpful to see a sample of the data. A graph could be included to facilitate interpretation.
4. Discussion:
- It is necessary to indicate what this research contributes with respect to other studies that have addressed the same topic, such as the case of “Medeiros, E., Guillermo Ramírez, M., Brustia, G., Dellagiacoma, A. C., & Mullan, C. A. (2023)”.
5. Conclusions:
- No additional solutions are proposed. Any future research lines?
Here are some additional notes on the translation:
- I have used the word "original" in the general comment to convey the idea that the study could have made a more significant contribution to the field if it had provided more in-depth analysis or insights.
- In the section on the introduction and background, I have clarified that the new regionalism could refer to a global phenomenon or a more specific European development. I have also added a reference to a relevant study that the authors could have cited.
- In the section on materials and methods, I have clarified that the authors should specify whether they used a quantitative or qualitative approach to their analysis. I have also suggested that they could have provided more information about the cases they studied.
- In the section on results, I have suggested that the authors could have improved the clarity and usefulness of Figure 3 by adding a legend and labels. I have also suggested that they could have provided a sample of the data they used to quantify the results.
- In the section on discussion, I have suggested that the authors could have more clearly explained how their research contributes to the existing literature. I have also suggested that they could have proposed some additional solutions or future research directions.
Author Response
Thank you very much for taking the time to review this manuscript. Please find the detailed responses below and the corresponding revisions/corrections highlighted/in track changes in the re-submitted files.
Comments and Suggestions for Authors
General information: The source or authorship of the figures should be indicated.
We filled in all the missing sources and authorship of the figures and tables.
Abstract: Clear and concise. The meaning of the acronym AEBR should be expressed.
We have explained the acronym AEBR and the other acronyms in the abstract (page 1, lines 13, 16, 22, 26).
- Introduction and background:
- Lines 39-41 discuss the beginning of the new regionalism. Is this worldwide? Or more specifically in Europe, within the framework of the European Economic Community?
- We have, accordingly, added more information to clarify this point, emphasizing the global nature of the phenomenon, its facilitators, and, in the same time, the specificity of the European Union, seen as a model of this new regionalism. We have also added more relevant references to support this statement (page 2, lines 49-62).
- A brief definition of the territorial scale of the mesoregional and macroregional levels would be helpful. Perhaps a footnote would suffice.
- We agree with this comment. Therefore, we have shortly defined the mesoregional and macroregional levels, by inserting the information in parentheses (page 2, lines 67-70).
- More background on innovation in cross-border regional systems is needed. For example: Makkonen, T., Weidenfeld, A., & Williams, A. M. (2017). Cross‐border regional innovation system integration: An analytical framework. Tijdschrift voor economische en sociale geografie, 108(6), 805-820.
- Thank you for your suggestion. We have read and included the recommended paper in the theoretic background (Page 3, line 126 - Reference 45).
- Materials and methods:
- It is essential to clarify some things in this section. Was any tool used to analyze the information? Is it a content analysis? What measures were taken to ensure the validity and reliability of the qualitative results?
- We have used content analysis to analyze the data collected, therefore our research is a qualitative one.
- It should also be made clear which cases were worked with. Generating a map would facilitate the task.
- Figure 3 shows the map of the 24 cases analyzed for institutional cooperation.
- Results:
- The cartography of Figure 3 needs improvement. A legend should be included that explains the elements incorporated (patches and points), and labels for the countries and regions of interest should be added. In addition, a legend and geographic north should be incorporated.
- We have added the legend to figure 3. Geographical north has also been indicated. We do not think it is necessary to label the EU Member States, as they are very well known.
- If reference is made to a quantification of the results by thematic categories (314-316), it would be helpful to see a sample of the data. A graph could be included to facilitate interpretation.
- We have additionally introduced the Graph 1 to better argue the statement and facilitate the understanding (former lines 314-316, now the lines 324-328)
- Discussion:
- It is necessary to indicate what this research contributes with respect to other studies that have addressed the same topic, such as the case of “Medeiros, E., Guillermo Ramírez, M., Brustia, G., Dellagiacoma, A. C., & Mullan, C. A. (2023)”.
- We have added arguments highlighting our contribution with respect to other studies (lines 408 – 414; 488-503)
- Conclusions:
- No additional solutions are proposed. Any future research lines?
- We have completed the study with information related to future research direction (lines 504-509)
Reviewer 2 Report
Comments and Suggestions for Authors
I read with interest the paper on the EU policies towards border regions within the EU.
Here are my suggestions for improvement, then I will be happy to review the paper again once the authors have revised this version.
General comments:
- Can you please better discuss the relevant literature, its debates, and the original contribution of this paper? What are the gaps you are trying to address? Why does it matter? How does it relate to the literature on ENP and on the EU Green deal? This should be discussed explicitly
- Explicitly state the research question, and better guide the reader throughout the paper and the research.
Specific comments:
- Title should never be capitalised, only the first letter
- The title is not clear, please rephrase it (no need to mention the programme)
- Never use contractions in the abstract. First time you use it, you should spell it our (EC, EU, etc)
- Avoid unsubstantiated sentences, such as the first one of the introduction; add a reference.
- Second paragraph: '80... it should be 1980s.
- My suggestion is for you to read and mention also the work of Serena Sandri on the European Green Deal (published in Mediterranean Politics journal) in which she discusses implications for internal and external countries (so border regions of the EU, including the ENP). I think it would be important to include it.
Comments on tables and figures:
- Table 1: fix the format, it would be better to have the boxes and lines
Author Response
Thank you very much for taking the time to review this manuscript. Please find the detailed responses below and the corresponding revisions/corrections highlighted/in track changes in the re-submitted files.
Comments and Suggestions for Authors
I read with interest the paper on the EU policies towards border regions within the EU. Here are my suggestions for improvement, then I will be happy to review the paper again once the authors have revised this version.
General comments:
- Can you please better discuss the relevant literature, its debates, and the original contribution of this paper? What are the gaps you are trying to address? Why does it matter? How does it relate to the literature on ENP and on the EU Green deal? This should be discussed explicitly.
Thank you for pointing this out. Therefore, we have introduced the debates from the specialty literature, discussing the different directions in the study of the regionalization, which differ by the way specialists are considering the relationship between regionalization and globalization (page 1, lines 37-45). All the statements are supported by relevant references. Furthermore, we have added more information to clarify the concept of „new regionalism”, emphasizing the global nature of the phenomenon, its facilitators, and, at the same time, the specificity of the European Union, seen as a model of this new regionalism. We have also added more relevant references to support this statement (page 2, lines 49-62).
We have clarified the contribution of the paper, the gaps we are addressing and their relevance in the section of Conclusions (page 15, lines 488-509).
- Explicitly state the research question, and better guide the reader throughout the paper and the research.
We agree with this comment. We have, accordingly, explicitly formulated the research hypothesis (page 4, lines 152-153).
To better guide the reader throughout the paper and the research, we have added a paragraph containing the way of organizing the paper and the research architecture (page 4, lines 164-174).
Specific comments:
- Title should never be capitalised, only the first letter
- The title is not clear, please rephrase it (no need to mention the programme)
Agree. We have made the requested modifications (page 1, lines 2-3).
The new title: „European Union tools for the sustainable development of border regions”
- Never use contractions in the abstract. First time you use it, you should spell it our (EC, EU, etc)
Agree. We have eliminated the acronyms in the abstract (page 1, lines 13, 16, 22, 26).
- Avoid unsubstantiated sentences, such as the first one of the introduction; add a reference.
Thank you for pointing this out. We have clarified the first paragraph and we have introduced references to support the ideas formulated there (page 1, lines 31-36).
- Second paragraph: '80... it should be 1980s.
Agree. We have operated the change (page 2, line 50).
- My suggestion is for you to read and mention also the work of Serena Sandri on the European Green Deal (published in Mediterranean Politics journal) in which she discusses implications for internal and external countries (so border regions of the EU, including the ENP). I think it would be important to include it.
Thank you for your suggestion. We have read and included the recommended paper in our study (Page 9, line 278-279; Reference 59).
Comments on tables and figures:
- Table 1: fix the format, it would be better to have the boxes and lines
Agree. We have operated the change (Table 1, pages 6-8).
Round 2
Reviewer 1 Report
Comments and Suggestions for Authors
Dear authors:
The quality of the paper has increased considerably. However, I suggest that some improvements be made before publication:
-The map in figure 3 should also include a scale and, I insist, the labels of at least the regions studied.
-The methodology could be more detailed, it is a qualitative procedure, but how did you do the data processing?
Author Response
-The map in figure 3 should also include a scale and, I insist, the labels of at least the regions studied.
Thank you for your recommendation. We have introduced graphic scaling to improve the image and labels indicating the borders where the obstacles were found.
However, in the article the map is only indicative for understanding the results, and especially the spatial distribution of the 24 proposals accepted for analysis in the direction of institutional cooperation.
-The methodology could be more detailed, it is a qualitative procedure, but how did you do the data processing?
Thank you for your comment. We have added a text that will further clarify the methodology (page 5, lines 196-202, the text marked in yellow).
We consider that the method presented synthetically covers the necessary elements of an exploratory analysis, generating hypotheses or causal relationships. Qualitative research does not require statistical analysis by definition.
Reviewer 2 Report
Comments and Suggestions for Authors
Good contribution to the literature
Author Response
Thank you again for your recommendations. They were very useful.